# Validity and Reliability of the Korean Version of the Acceptance and Action Questionnaire-Stigma (AAQ-S-K)

**DOI:** 10.3390/healthcare9101355

**Published:** 2021-10-12

**Authors:** Hyunjin Lee, Myoungjin Kwon, Kawoun Seo

**Affiliations:** 1Department of Nursing, College of Nursing, Eulji University, Daejeon 34824, Korea; hjlee0815@daum.net; 2Department of Nursing, Daejeon University, Daejeon 34520, Korea; 3Department of Nursing, Joongbu University, Goyang 32713, Korea

**Keywords:** acceptance and commitment therapy, diabetes mellitus, social stigma, Acceptance and Action Questionnaire-Stigma (AAQ-S)

## Abstract

This study investigated the validity and reliability of the Korean version of the Acceptance and Action Questionnaire-Stigma (AAQ-S-K) in people with diabetes. A total of 208 patients with diabetes participated in the study. After performing forward and backward translation of the original version of the AAQ-S into Korean, its validity (construct and concurrent validity) and reliability were assessed. Construct validity measured using confirmatory factor analysis showed a good fit. Concurrent validity was confirmed through the significant correlation between the AAQ-S-K, acceptance and action, diabetes self-stigma and diabetes distress. The AAQ-S-K was positively correlated with acceptance and action and had a negative correlation with diabetes self-stigma and diabetes distress. The reliability of the AAQ-S-K ranged from 0.82 to 0.88. The AAQ-S-K can be applied to assess stigma acceptance and action in Korean patients with diabetes and to compare the level of psychological flexibility of patients with diabetes internationally.

## 1. Introduction

Psychological flexibility is a positive concept that helps people continue to act in a direction to achieve their goals and values by actively accepting their current personal experiences through a flexible process of perception [1]. In other words, it means having an attitude of acceptance and willingness to experience negative and harmful consequences [2,3]. A recent analysis of patients with mental health problems reported that the enhancement of psychological flexibility reduces pathological problems and improves psychological health [4,5].

Psychological flexibility is the ability to commit to a worthy activity pattern, which is derived from acceptance and commitment therapy (ACT), an approach that includes six processes: acceptance, defusion, self, commitment, the now and values [6]. Recently, it was found that ACT reduces negative self-perception and stigmatization in the process of accepting the existence of diseases in individuals with health problems [3] and brings about meaningful changes in their clinical outcomes [7,8]. Particularly, patients with diabetes, a representative chronic disease, have a negative perception of themselves, which leads to self-stigma [9]. This, in turn, reduces psychological flexibility and results in inappropriate self-management due to psychological problems, such as depression [10,11].

Self-stigma refers to recognizing negative attitudes or stereotypes about one’s own health condition as a fact and applying them specifically to oneself, indicating real or imaginary fear of discrimination and shame, thereby inducing complications, barriers and burden in patients with chronic diseases [12,13,14,15]. Self-stigma can also occur in patients with diabetes [16,17,18]. Therefore, assessing and managing the degree of stigma acceptance behavior in patients with diabetes is important for effective diabetes management.

ACT for patients with diabetes increases their psychological flexibility, helping them accept unpleasant thoughts and assumptions related to diabetes and improve their quality of life by devoting themselves to their values [19]. It can also improve self-efficacy and perceptual stress in patients with diabetes [20]. Based on these findings, measuring psychological flexibility of patients with diabetes is thought to be helpful in reducing negative perceptions of the disease and improving coping strategies [20]. However, despite these promising findings, there are no tools available in Korea to evaluate the stigma acceptance behavior of patients with diabetes.

The Acceptance and Action Questionnaire-Stigma (AAQ-S), developed by Levin et al. [21], is one such tool that evaluates acceptance behavior toward stigma. It measures psychological flexibility with stigmatizing thoughts and the validity and reliability of the tool have been demonstrated. To use this tool in a Korean setting, it is necessary to evaluate whether the tool is appropriate for measuring the acceptance of stigmatizing thoughts in Korean patients with diabetes. Therefore, this study aimed to evaluate the validity and reliability of the Korean version of the AAQ-S (AAQ-S-K) by applying it to Korean patients with diabetes. This study is expected to provide an effective evaluation tool for acceptance and commitment-based interventions and research focused on Korean patients with diabetes.

## 2. Materials and Methods

### 2.1. Design and Participants

This was a descriptive cross-sectional study. The survey was collected from 26 March to 28 March 2020, through a survey institution (PMI Co., Ltd., Seoul, Korea) for patients diagnosed with diabetes who understood the purpose of the study and voluntarily agreed to participate in the study. A self-report questionnaire consisting of general characteristics, acceptance and action questionnaire-stigma, acceptance and action, diabetes self-stigma and diabetes distress was prepared as an online questionnaire and distributed to 4300 panelists of the survey institution (PMI). The sample size required to perform confirmatory factor analysis (CFA) to verify construct validity was at least 150 [22], but the questionnaire was conducted with 250 diabetes patients in consideration of response fidelity. The questionnaire was conducted for those who answered that they had diabetes in both questions of current health problems and diseases diagnosed by a doctor among a total of 15 chronic diseases. For participants who did not select diabetes for either of the two questions, the questionnaire was set to automatically end. Afterwards, they were asked to answer 5 questions included in diabetes-related characteristics and, after answering the diabetes-related questions, the rest of the questionnaire was composed. After 250 responses were collected on a first-come, first-served basis, the survey was ended. The final analysis included 208 participants, excluding data from 42 who had insufficient responses.

### 2.2. Measurements

The questionnaire used in this study consisted of 15 items of general characteristics, 21 items of acceptance and action questionnaire-stigma, 16 items of acceptance and action, 16 items of 16 diabetes self-stigma scale and 20 items of diabetes stress questions.

#### 2.2.1. Acceptance and Action Questionnaire-Stigma

The AAQ-S was translated into Korean using the translation-reverse translation process with the permission of the original developers [21]. The AAQ-S consists of a total of 21 items, including 11 items in the “Psychological Inflexibility Subscale” and 10 items in the “Psychological Flexibility Subscale.” The translated tool was corrected through face validity. The AAQ-S consists of 21 items evaluated on a 7-point Likert scale ranging from 1 = “not at all” to 7 = “always.” The Cronbach’s α of the original tool was 0.82.

#### 2.2.2. Acceptance and Action

The Korean version of the Acceptance and Action Questionnaire, which was originally developed by Hayes et al. [23] and adapted in Korean by Moon [24], was used. It measures the degree to which one is willing to accept a thought or emotion, acting in a way that is consistent with one’s values and goals. The scale consists of 16 items rated on a 7-point Likert scale ranging from 1 = “not at all” to 7 = “always.” A higher total score indicates higher degree of acceptance. The Cronbach’s α was 0.82 in Moon [24] and 0.77 in this study.

#### 2.2.3. Diabetes Self-Stigma

The Diabetes Self-Stigma Scale developed by Seo and Song [25] was used to measure self-stigma. The tool consists of 16 items in four sub-domains: comparative inability, social withdrawal, self-devaluation and apprehensive feeling. All items are measured on a 5-point Likert scale (1 = “not at all”, 2 = “not at all”, 3 = “average”, 4 = “yes”, 5 = “very much”). The Cronbach’s α was 0.89 in the original study [25] and 0.93 in this study.

#### 2.2.4. Diabetes Distress

To measure stress in patients with diabetes, Polonsky et al. [26] translated and secured the Problem Areas in Diabetes (PAID) scale according to the Korean situation using the Korean version of the scale (PAID-K) [27]. The tool consists of 20 items rated on a 5-point Likert scale ranging from 1 to 5, indicating from no problem to a very serious problem. A higher score indicates higher level of perceived stress. The Cronbach’s α was 0.95 in the Korean study [27] and 0.94 in this study.

### 2.3. Procedure

The double translation method suggested by Waltz et al. [28] was used to translate the AAQ-S into Korean. One nursing professor and one nursing doctor, who were bilingual speakers of Korean and English, translated the AAQ-S into Korean and revised and supplemented the translated version while comparing the results. Another doctor who spoke English as his mother tongue and was fluent in Korean translated the Korean version back into English. After the reverse translation, a nursing major fluent in English and Korean performed a comparative analysis and verified whether there were any items with differences in meaning from the original tool. To check the content validity of the translated tool, two nursing professors and three diabetes experts verified the validity of the questionnaire. They also checked whether the contents of this questionnaire were applicable to Korean culture. The content validity index (CVI) evaluates the degree to which the tool is appropriate for the measurement of stigma acceptance behavior on a 4-point Likert scale ranging from 4 = “strongly agree” to 1 = “strongly disagree.” All items were confirmed to be valid with a CVI of 0.8 or higher. In addition, it was judged that the contents of the questionnaire can be applied to Korean culture.

Before the validity test, the translated questionnaire was administered to 10 patients with diabetes and the time taken to fill out the questionnaire and the responses of the participants were observed. Additionally, the participants were asked to present their opinions when the meaning was unclear or they did not understand vocabulary or sentences while filling out the questionnaire. The time to respond to the questionnaire ranged between 3 and 5 min. The final translation of the tool was completed without any modifications, as there were no complaints of difficulty in responding.

Then, the construct and concurrent validity of the AAQ-S-K was verified. For construct validity, sub-concepts were derived at the time of development of the original tool and CFA, as well as item convergent and discriminant validation, was performed on the basis that CFA was more appropriate than exploratory factor analysis [29]. Convergence validity can be evaluated through composite or construct reliability because there must be a high correlation between values measured by different methods to evaluate a given concept. Concept reliability measures the internal consistency of items in a tool. Discriminant validity can be evaluated through the average variance extracted (AVE), as there must be a clear difference in measured values between different variables. In relation to the measurement error, it evaluates the size of the variance obtained from the entire instrument. It has a strict standard of 0.5 or more and, if the square of the correlation coefficient between each variable is lower than the AVE, it is judged that there is discriminant validity. Concurrent validity was assessed through the Pearson correlation analysis of four variables by simultaneously applying the acceptance behavior questionnaire to measure acceptance and action and the diabetes self-stigma and diabetes distress questionnaires, which were found to be related to stigma-accepting behavior in previous studies. The total score of each tool was evaluated by testing whether there was a correlation. Reliability of the tool was verified by calculating Cronbach’s α, which indicates internal consistency.

### 2.4. Data Analysis

The data were analyzed using SPSS/WIN (IBM, Armonk, NY, USA) version 24.0 and Amos (IBM, Armonk, NY, USA) version 22.0. The general characteristics of the participants were analyzed using descriptive statistics and CFA was performed to verify model suitability between the sub-domain structures of the existing items to verify construct validity. The goodness-of fit-index (GFI), root mean square residual (RMR) and root mean square error of approximation (RMSEA) were used as model fit indices. The comparative fit index (CFI), Tucker–Lewis index (TLI) and incremental fit index (IFI) were identified as incremental fit indices. The convergence validity of the items was verified through standardized factor loading, concept reliability and mean variance extraction (AVE) results. The discriminant validity of the items was confirmed using the correlation coefficients and AVE values. The concurrent validity was verified using the Pearson correlation analysis and reliability was calculated as Cronbach’s α.

## 3. Results

### 3.1. Participants’ Characteristics

The majority of the participants in this study were male (80.8%) and the average age was 50.67 (±10.9) years. Approximately 74.5% had university education and those with spouses and jobs accounted for 75.4% and 79.8%, respectively. Regarding social activities, the majority of those who were active less than once a month (40.4%) reported an average subjective health status (49.5%). In terms of diabetes-related characteristics, the average duration of the disease was 6.38 (±6.49) years and the majority of the participants were treated at general clinics (43.3%). As for the type of treatment, those taking oral hypoglycemic agents accounted for the majority (59.1%) and only 6.3% underwent insulin therapy. About 40.9% had experience in diabetes-related education.

### 3.2. Item Analysis

For the item analysis, the mean and standard deviation of each item and each factor were measured and normality was evaluated by checking skewness and kurtosis. The average item score was 3.83–4.61, the standard deviation was 1.13–1.40 and the average and standard deviation of the total score was 89.33 ± 17.06. Skewness and kurtosis were evaluated as satisfying normality [30] when the standard value was less than ±1.97 at the 5% significance level. The correlation coefficient evaluating the correlation between individual questions and total score was interpreted as showing a low correlation when the value was less than 0.30 [31]; the correlation coefficients in this study ranged from 0.53 to 0.70. The ceiling and floor effects refer to the frequency (%) of the highest and lowest scores of each item. Referring to Kane [32], all the items in this study had a frequency less than 30%.

### 3.3. Confirmatory Factor Analysis

The construct validity of the AAQ-S-K was verified through CFA. The standardization coefficients of the items corresponding to each factor were all above 0.50, confirming the validity of the items. Accordingly, the suitability of the AAQ-S-K, consisting of 2 factors and 21 items, was confirmed. The absolute fit index of χ^2^ = 471.35 (*p* < 0.001), the degree of freedom (df) = 188 and Normed χ^2^ (χ^2^/df) = 2.50. RMR was 0.08 and RMSEA was 0.08. Referring to Roh [33], Normed χ^2^ should be less than 3 and RMR and RMSEA range from 0.05 to 0.08 or are less in value. All incremental fit indices above 0.90 are accepted as good fit [33]. The results of this study showed that CFI and IFI had a good fit at 0.90 or higher, but GFI and TLI did not meet the acceptance criteria (Table 1).

### 3.4. Convergence and Discriminant Validity

Concept reliability and mean variance extraction were evaluated to verify the convergent and discriminant validity of the AAQ-S-K (Table 2). In this study, the concept reliability was between 0.97 and 0.98, which was above the allowable limit of 0.70; therefore, the convergent validity was secured. The AVE value ranged from 0.81 to 0.82, which was above the standard value of 0.50 and was larger than the square of the correlation of each variable, which was 0.64, ensuring discriminant validity.

### 3.5. Concurrent Validity

The AAQ-S-K showed a positive correlation with acceptance and action (*r* = 0.22, *p* = 0.002) and a negative correlation with diabetes self-stigma (*r* = −0.34, *p* < 0.001) and diabetes distress (*r* = −0.29, *p* < 0.001).

## 4. Discussion

This study was conducted to verify the validity and reliability of the AAQ-S-K which measures the acceptance of stigmatizing thoughts in Korean patients with diabetes. The AAQ-S-K consists of 21 items, grouped into two sub-factors. The results of CFA conducted to confirm the construct validity of the AAQ-S-K showed that the standardization coefficients of all items corresponding to each factor were 0.50 or higher, confirming that the items were valid for the corresponding factor. Additionally, as a result of evaluating the factor structure of the tool and model fit for the items, χ^2^/df was found to be ≤3.0, RMR and RMSEA were ≤0.08 and CFI and IFI were ≥0.90, indicating excellent fit. GFI and TLI did not meet the acceptance criteria. These results are thought to be related to questions 5, 11 and 19, which had relatively low beta values. These questions were poorly understood by the participants at the time of the survey and should be revised and improved to make it easier for the participants to understand. Furthermore, it is difficult to compare the findings of this study with other studies because the original tool does not present the model fit results. However, in a study by Trigueros et al. [34], RMSEA, CFI and IFI were all found to meet the criteria. This might be because the understanding of the tool varies depending on the target; therefore, in order to further improve the validity of the AAQ-S-K, revision and improvements should be made to increase the understanding of the tool.

The first sub-factor, factor 1, consists of 11 items and measures psychological inflexibility with stigmatizing thoughts, whereas factor 2, consisting of 10 items, evaluates psychological flexibility with stigmatizing thoughts. The convergent and discriminant validity of these sub-domains was confirmed, which proves that the sub-domains of the AAQ-S-K are distinguished for measuring psychological flexibility and inflexibility and are related to each other. In general, the measurement tools of AAQ focus on measuring experience avoidance and psychological rigidity. However, since more importance is being given to the broader concept of psychological flexibility rather than experience avoidance [35,36], it is desirable to measure psychological flexibility and rigidity together. As the AAQ-S-K measures both psychological flexibility and rigidity with stigmatizing thoughts of patients with diabetes, it is expected to be helpful in measuring their stigma acceptance behavior. With this questionnaire, healthcare providers can determine the flexibility or rigidity of stigma thoughts that people with diabetes have. Through this confirmation, it is possible to estimate how flexibly they can cope with the self-stigma that affects diabetes patients. This flexibility for self-stigma will help diabetes patients, even if they have self-stigma, so that it does not affect their self-care. However, currently, there are few studies conducted on this area. Efforts are needed to measure the degree of flexibility for stigma in diabetes patients in the future.

To verify the concurrent validity of the AAQ-S-K, the correlation among acceptance and action, diabetes self-stigma and diabetes distress was confirmed. A significant positive correlation was found with acceptance and action and the relationship between diabetes self-stigma and diabetes distress was also positive. This means that the higher the AAQ-S-K score, the higher the general acceptance and action, which indicates that the AAQ-S-K is appropriate for measuring acceptance and action. Additionally, the higher the AAQ-S-K score, the lower the diabetes self-stigma and diabetes distress. This finding is consistent with the results obtained by Seo [37], which reported less diabetes stress and occurrence of self-stigma when patients with diabetes accepted the existence of the disease. Thus, the AAQ-S-K is appropriate to measure the acceptance of stigma in patients with diabetes. However, the relationship between the AAQ-S-K, diabetes self-stigma and diabetes distress remains unclear. Therefore, future studies should focus on investigating the relationship among diabetes self-stigma, diabetes distress and diabetes acceptance.

The Cronbach’s α for all 21 items was 0.93. Additionally, the Cronbach’s α for the psychological inflexibility sub-factor was 0.88 and for psychological flexibility was 0.89, indicating a high level of internal consistency. These findings conform with Trigueros et al.’s [34], for which Cronbach’s α for all items was 0.96 and it was 70 or higher, which is the standard suggested by Nunnally and Bernstein [38]. This indicates the reliability of the AAQ-S-K for measuring the stigma acceptance behavior of Korean patients with diabetes.

The significance of the AAQ-S-K is presented based on the validation results. First, this tool objectively measures the psychological flexibility of patients with diabetes by measuring their acceptance of stigmatizing thoughts. Second, it is possible to compare the AAQ-S-K with the AAQ-S translated into other languages. Third, this tool focuses on the emotional aspects of being diagnosed with diabetes and can be applied in various studies in the future. Despite these advantages, there are some limitations to this study. First, the study sample was rather small. In addition, there is a possibility that bias may have occurred because some participants were included. Therefore, care should be taken when generalizing the results of the study. Second, in the construct validity analysis, some of the fit indices did not meet the criteria. To address these limitations, further studies need to validate items using a larger study sample through random sampling. These studies will help to measure the acceptance of stigma thoughts in diabetes patients.

## 5. Conclusions

This study confirmed the construct validity of the AAQ-S-K, which consists of 21 items grouped into two sub-factors, through CFA and the concurrent validity was verified by confirming the correlation with related concepts. Additionally, the reliability of the tool was guaranteed owing to the high internal consistency of the items. These findings indicate that the AAQ-S-K can be used to measure the acceptance of stigmatizing thoughts in patients with diabetes. Based on the results of this study, we suggest the following: First, future studies should objectively measure the degree of acceptance of diabetes stigma in patients with diabetes using the AAQ-S-K. Second, interventions using the AAQ-S-K should be implemented to increase the psychological flexibility of patients with diabetes.

## Figures and Tables

**Table 1 healthcare-09-01355-t001:** Summary of fit indices from confirmatory factor analysis (*N* = 208).

Variables	χ^2^/DF	GFI	RMR	RMSEA	CFI	TLI	IFI
Evaluation criteria	≤3	≥0.90	≤0.05~0.08	≤0.05~0.08	≥0.90	≥0.90	≥0.90
AAQ-S-K	2.50	0.86	0.08	0.08	0.90	0.87	0.90

DF = degree of freedom; RMR = root mean-square residual; RMSEA = root mean square error of approximation; GFI = goodness of fit index; CFI = comparative fit index; TLI = Tucker-Lewis index; IFI = incremental fit index; AAQ-S-K = Korean version of the Acceptance and Action Questionnaire-Stigma.

**Table 2 healthcare-09-01355-t002:** Convergent validity of the Korean version of the Acceptance and Action Questionnaire-Stigma Scale (*N* = 208).

Factors	Items	Estimate	Standardized Estimates	Standardized Error	Critical Ratio	*p*	AVE	CR	Cronbach’s α
Psychological Inflexibility	Q1	1.00	0.71				0.81	0.98	0.88
	Q2	0.94	0.61	0.11	8.43	<0.001			
	Q3	1.05	0.61	0.12	8.44	<0.001			
	Q4	0.96	0.65	0.10	8.92	<0.001			
	Q5	0.92	0.58	0.11	8.06	<0.001			
	Q6	0.81	0.60	0.09	8.25	<0.001			
	Q7	1.10	0.75	0.10	10.41	<0.001			
	Q8	1.04	0.72	0.10	9.97	<0.001			
	Q9	0.94	0.67	0.10	9.20	<0.001			
	Q10	0.99	0.66	0.10	9.10	<0.001			
	Q11	0.97	0.59	0.12	8.17	<0.001			
* Psychological Flexibility	Q12	1.00	0.64				0.82	0.97	0.89
	Q13	0.98	0.64	0.12	8.09	<0.001			
	Q14	0.94	0.66	0.11	8.34	<0.001			
	Q15	1.06	0.72	0.11	8.94	<0.001			
	Q16	1.08	0.72	0.12	8.89	<0.001			
	Q17	0.94	0.71	0.10	8.78	<0.001			
	Q18	0.92	0.66	0.11	8.34	<0.001			
	Q19	0.76	0.56	0.11	7.24	<0.001			
	Q20	1.04	0.72	0.11	8.95	<0.001			
	Q21	1.00	0.72	0.10	8.90	<0.001			
Overall									0.93

AVE = average variance extracted; CR = concept reliability; * Psychological flexibility is an inverse question.

## Data Availability

The data presented in this study are available upon request from the corresponding author. The data are not publicly available because of privacy concerns.

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
