# Peer review of "Validity and Reliability of the Korean Version of the Acceptance and Action Questionnaire-Stigma (AAQ-S-K)"

_healthcare, 2021, doi:10.3390/healthcare9101355_

Round 1
Reviewer 1 Report
I do not have the statistical expertise to understand and assist with the presentation of data. I do hope another reviewer with this expertise can better evaluate the quality of the work and offer suggestions. I am not familiar with confirmatory factor analysis or convergence and discriminant validity. The authors need to think about whether they want to simplify the article in the effort to make their work accessible to a wider audience.
The authors have done a very good job with the double translation method. Clearly, a strong effort was made here.
I found the sample of a good size and appreciate the authors' discussion that a larger sample size will be a benefit in the future.
The conclusion is weak and too general. Obviously, considerable time and effort were put into this work. I'd like to see the authors be more specific about their thoughts on what can be done to reduce self-stigma among Koreans with diabetes.
Author Response
Response to Reviewer 1 Comments
We would like to thank the reviewers for their encouragement and thorough critique. As required, we have addressed the reviewers’ comments below point-by-point. In addition, we have also made revisions to the manuscript, as advised; these changes have been marked with red font in the main text for clarity.
Point 1: I do not have the statistical expertise to understand and assist with the presentation of data. I do hope another reviewer with this expertise can better evaluate the quality of the work and offer suggestions. I am not familiar with confirmatory factor analysis or convergence and discriminant validity. The authors need to think about whether they want to simplify the article in the effort to make their work accessible to a wider audience.
The authors have done a very good job with the double translation method. Clearly, a strong effort was made here.
I found the sample of a good size and appreciate the authors' discussion that a larger sample size will be a benefit in the future.
The conclusion is weak and too general. Obviously, considerable time and effort were put into this work. I'd like to see the authors be more specific about their thoughts on what can be done to reduce self-stigma among Koreans with diabetes.
Response 1: Reflecting the reviewer's comments, the following content has been added to the discussion section.
“The first sub-factor, factor 1, consists of 11 items and measures psychological inflexibility with stigmatizing thoughts, whereas factor 2, consisting of 10 items, evaluates psychological flexibility with stigmatizing thoughts. The convergent and discriminant validity of these sub-domains was confirmed, which proves that the sub-domains of the AAQ-S-K are distinguished for measuring psychological flexibility and inflexibility and are related to each other. In general, the measurement tools of AAQ focus on measuring experience avoidance and psychological rigidity. However, since more importance is being given to the broader concept of psychological flexibility rather than experience avoidance [35,36], it is desirable to measure psychological flexibility and rigidity together. As the AAQ-S-K measures both psychological flexibility and rigidity with stigmatizing thoughts of patients with diabetes, it is expected to be helpful in measuring their stigma acceptance behavior. With this questannaire, heatlhcare providers can determine the flexibility or rigidity of stigma thoughts that people with diabetes have. Through this confirmation, it is possible to estimate how flexibly they can cope with the self-stigma that diabetes patients. This flexibility for self-stigma will help diabetes patients, even if they have self-stigma, so that it does not affect their self-care. However, currently there are few studies conducted on this area. Efforts are needed to measure the degree of flexibility for stigma in diabetes patients in the future.” (Page 7)

Reviewer 2 Report
The manuscript entitled "Validity and Reliability of the Korean Version of the Acceptance and Action Questionnaire-Stigma (AAQ-S-K)" is a well-written article. The validation method is very good and the results are clearly reported. Thank you for the opportunity to read this interesting work. However, some suggested changes in the Method section may improve the content.
- Authors wrote (page 2, lines 69-70): "A total of 250 participants aged 20 years or older were selected from among 4,300 patients". Please add the inclusion and exclusion criteria.
- How was the survey distributed (online or otherwise?). If stationery, where it was performed (e.g., at home or in a hospital - in a corridor, physician room, etc.)? How many hospitals were involved in the study? Who distributed it (nurses or physicians?). What were the circumstances (e.g. during a medical visit)? Please, describe more details to allow the replication of this study. What form was the questionnaire (online, e.g. Google forms; or paper-and-pencil)?
- How many missing values were found? How did you dealt with it?
- Could you provide sample items for each questionnaire in the method section?
- Please provide the names of the original AAQ-S subscales and how many items are included in each of them.
- It is suggested to use the symbol χ2 instead of CMIN or "Chi-square", consistently in the whole results section (including tables).
- Table 3 is redundant as all results of correlation analysis are described in the text. However, it would be valuable to perform multiple linear regression analysis for the AAQ-S-K as dependent variable, and acceptance and action, self-stigma, and diabetes distress as predictors.
Author Response
Response to Reviewer 2 Comments
We would like to thank the reviewers for their encouragement and thorough critique. As required, we have addressed the reviewers’ comments below point-by-point. In addition, we have also made revisions to the manuscript, as advised; these changes have been marked with red font in the main text for clarity.
Point 1: Authors wrote (page 2, lines 69-70): "A total of 250 participants aged 20 years or older were selected from among 4,300 patients". Please add the inclusion and exclusion criteria.
Response 1: The contents of the participants selection process have been added as follows.
“This was a descriptive cross-sectional study. The survey was collected from March 26 to March 28, 2020, through a survey institution (PMI Co., Ltd.) for patients diagnosed with diabetes who understood the purpose of the study and voluntarily agreed to participate in the study. A self-report questionnaire consisting of general characteristics, acceptance and action questionnaire-stigma, acceptance and action, diabetes self-stigma, and diabetes distress was prepared as an online questionnaire and distributed to 4,300 panelists of the survey institution (PMI). The sample size required to perform confirmatory factor analysis (CFA) to verify construct validity was at least 150 [22], but the questionnaire was conducted with 250 diabetes patients in consideration of response fidelity. The questionnaire was conducted for those who answered that they had diabetes in both questions of current health problems and diseases diagnosed by a doctor among a total of 15 chronic diseases. For participants who did not select diabetes for either of the two questions, the questionnaire was set to automatically end. Afterwards, they were asked to answer 5 questions included in diabetes-related characteristics, and after answering the diabetes-related questions, the rest of the questionnaire was composed. After 250 responses were collected on a first-come, first-served basis, the survey was ended. The final analysis included 208 participants, excluding data from 42 who had insufficient responses.”(Page 2)
Point 2: How was the survey distributed (online or otherwise?). If stationery, where it was performed (e.g., at home or in a hospital - in a corridor, physician room, etc.)? How many hospitals were involved in the study? Who distributed it (nurses or physicians?). What were the circumstances (e.g. during a medical visit)? Please, describe more details to allow the replication of this study. What form was the questionnaire (online, e.g. Google forms; or paper-and-pencil)?
Response 2: The following explanation has been added for the data collection method.
“This was a descriptive cross-sectional study. The survey was collected from March 26 to March 28, 2020, through a survey institution (PMI Co., Ltd.) for patients diagnosed with diabetes who understood the purpose of the study and voluntarily agreed to participate in the study. A self-report questionnaire consisting of general characteristics, acceptance and action questionnaire-stigma, acceptance and action, diabetes self-stigma, and diabetes distress was prepared as an online questionnaire and distributed to 4,300 panelists of the survey institution (PMI). The sample size required to perform confirmatory factor analysis (CFA) to verify construct validity was at least 150 [22], but the questionnaire was conducted with 250 diabetes patients in consideration of response fidelity. The questionnaire was conducted for those who answered that they had diabetes in both questions of current health problems and diseases diagnosed by a doctor among a total of 15 chronic diseases. For participants who did not select diabetes for either of the two questions, the questionnaire was set to automatically end. Afterwards, they were asked to answer 5 questions included in diabetes-related characteristics, and after answering the diabetes-related questions, the rest of the questionnaire was composed. After 250 responses were collected on a first-come, first-served basis, the survey was ended. The final analysis included 208 participants, excluding data from 42 who had insufficient responses.” (Page 2)
Point 3: How many missing values were found? How did you dealt with it?
Response 3: Samples with missing data were removed from the final analysis. The following has been added to the text.
“The final analysis included 208 participants, excluding data from 42 who had insufficient responses.”(Page 2)
Point 4: Could you provide sample items for each questionnaire in the method section?
Response 4: Added the number of items for each question in the method section.
“ The questionnaire used in this study consisted of 15 items of general characteristics, 21 items of acceptance and action questionnaire-stigma, 16 items of acceptance and action , 16 items of 16 diabetes self-stigma scale and 20 items of diabetes stress questions.”(Page 2)
Point 5: Please provide the names of the original AAQ-S subscales and how many items are included in each of them.
Response 5: In the Measurement section, we provided the names of the subdomains of the original AAQ-S scale and the number of items included in each subscale.
“The AAQ-S was translated into Korean using the translation-reverse translation process with the permission of the original developers (Levin et al. [21]). The AAQ-S consists of a total of 21 items, including 11 items in “Psychological Inflexibility Subscale” and 10 items in “Psychological Flexibility Subscale.” The translated tool was corrected through face validity. The AAQ-S consists of 21 items evaluated on a 7-point Likert scale ranging from 1=“not at all” to 7=“always.” The Cronbach’s α of the original tool was .82.”(Page 2)
Point 6: It is suggested to use the symbol χ2 instead of CMIN or "Chi-square", consistently in the whole results section (including tables).
Response 6: We used χ2 instead of CMIN or Chi-square, including tables.
“The construct validity of the AAQ-S-K was verified through CFA. The standardization coefficients of the items corresponding to each factor were all above .50, confirming the validity of the items. Accordingly, the suitability of the AAQ-S-K, consisting of two factors and 21 items, was confirmed. The absolute fit index of χ2 =471.35 (p<.001), the degree of freedom (df)=188, and Normed χ2 (χ2 /df)=2.50. RMR was .08 and RMSEA was .08. Referring to Roh [33], Normed χ2 should be less than 3, and RMR and RMSEA range from .05 to .08 or are less in value. All incremental fit indices above .90 are accepted as good fit [33]. The results of this study showed that CFI and IFI had a good fit at .90 or higher, but GFI and TLI did not meet the acceptance criteria (Table 1). “(Pagge 4-5)
Table 1. Summary of fit indices from confirmatory factor analysis (N=208).
|
Variables |
χ2/DF |
GFI |
RMR |
RMSEA |
CFI |
TLI |
IFI |
|
Evaluation criteria |
≤3 |
≥.90 |
≤.05~.08 |
≤.05~.08 |
≥.90 |
≥.90 |
≥.90 |
|
AAQ-S-K |
2.50 |
.86 |
.08 |
.08 |
.90 |
.87 |
.90 |
|
DF=degree of freedom; RMR=root mean-square residual; RMSEA=root mean square error of approximation; GFI=goodness of fit Index; CFI=comparative fit index; TLI=Tucker-Lewis index; IFI=incremental fit index; AAQ-S-K=Korean version of the Acceptance and Action Questionnaire-Stigma |
|||||||
Point 7: Table 3 is redundant as all results of correlation analysis are described in the text. However, it would be valuable to perform multiple linear regression analysis for the AAQ-S-K as dependent variable, and acceptance and action, self-stigma, and diabetes distress as predictors.
Response 7: Table 3 has been deleted according to the reviewers' opinions. Multiple regression analysis will be conducted as a follow-up study. thank you.

Reviewer 3 Report
This is a very interesting research about the validity and reliability of the Korean Version of the AAQ-S-K questionnaire, and I really found it interesting, because the process of validation describe in the article is not only interesting, but also useful, for other researchers interested in this specific topic, or in the own validation process. Nevertheless, there are some minor aspects of the methodology that I humbly think that could be revised, to make some parts easier to understand, and better contextualize the conclusions. These are the aspects that I would propose to revise:
In Materials and Methods, Design and Participants, the authors write that «This was a descriptive cross-sectional study. The participants were patients diagnosed with diabetes who understood the purpose of the study and voluntarily agreed to participate in the study. A total of 250 participants aged 20 years or older were selected from among 4,300 patients». Selection bias is probably the most important limitation of this research. Selection bias is almost unavoidable, so the authors must make a considerable effort to clearly describe where they obtain the sample from, so the readers can have a clear idea of the main features of that sample, which also should be described. Therefore, I would propose the authors to better describe where the sample is obtained from.
In that same section, the authors write that «The required sample size was at least 150 to perform confirmatory factor analysis (CFA) to verify construct validity». It would interesting if they describe, in the article, if they performed any sample size estimation, and which method did they employ, in that case.
The authors also write that «the questionnaire was distributed to 250 patients with diabetes considering the recovery rate and response fidelity». Again, it is important to consider the potential selection bias of the research. To better understand the results (and therefore the conclusions), it would be very interesting to know, with more detail, how the patients where chosen, the attrition rate, or other factors related to the sample selection.
In Measurements, Acceptance and Action Questionnaire-Stigma, the authors write that «The AAQ-S was translated into Korean using the translation-reverse translation process with the permission of the original developers». Here, and in the next questionnaires described, it would be useful to know if the authors performed transcultural adaptation, or if it was not required (and why, in this case).
In the limitations section, perhaps the authors could make some kind of comment of the potential selection bias (if they consider that it could exist), and how it could affect their results and conclusions.
In think that if these aspects are better explained, the readers could better contextualize and better understand the results and the conclusions obtained. Nevertheless, I think that this is, overall, a very good research that can be useful for many researchers. Congratulations.
Author Response
Response to Reviewer 3 Comments
We would like to thank the reviewers for their encouragement and thorough critique. As required, we have addressed the reviewers’ comments below point-by-point. In addition, we have also made revisions to the manuscript, as advised; these changes have been marked with red font in the main text for clarity.
Point 1: In Materials and Methods, Design and Participants, the authors write that «This was a descriptive cross-sectional study. The participants were patients diagnosed with diabetes who understood the purpose of the study and voluntarily agreed to participate in the study. A total of 250 participants aged 20 years or older were selected from among 4,300 patients». Selection bias is probably the most important limitation of this research. Selection bias is almost unavoidable, so the authors must make a considerable effort to clearly describe where they obtain the sample from, so the readers can have a clear idea of the main features of that sample, which also should be described. Therefore, I would propose the authors to better describe where the sample is obtained from.
In that same section, the authors write that «The required sample size was at least 150 to perform confirmatory factor analysis (CFA) to verify construct validity». It would interesting if they describe, in the article, if they performed any sample size estimation, and which method did they employ, in that case.
The authors also write that «the questionnaire was distributed to 250 patients with diabetes considering the recovery rate and response fidelity». Again, it is important to consider the potential selection bias of the research. To better understand the results (and therefore the conclusions), it would be very interesting to know, with more detail, how the patients where chosen, the attrition rate, or other factors related to the sample selection.
Response 1: Reflecting the opinions of reviewers, the following information has been added to the method section.
“This was a descriptive cross-sectional study. The survey was collected from March 26 to March 28, 2020, through a survey institution (PMI Co., Ltd.) for patients diagnosed with diabetes who understood the purpose of the study and voluntarily agreed to participate in the study. A self-report questionnaire consisting of general characteristics, acceptance and action questionnaire-stigma, acceptance and action, diabetes self-stigma, and diabetes distress was prepared as an online questionnaire and distributed to 4,300 panelists of the survey institution (PMI). The sample size required to perform confirmatory factor analysis (CFA) to verify construct validity was at least 150 [22], but the questionnaire was conducted with 250 diabetes patients in consideration of response fidelity. The questionnaire was conducted for those who answered that they had diabetes in both questions of current health problems and diseases diagnosed by a doctor among a total of 15 chronic diseases. For participants who did not select diabetes for either of the two questions, the questionnaire was set to automatically end. Afterwards, they were asked to answer 5 questions included in diabetes-related characteristics, and after answering the diabetes-related questions, the rest of the questionnaire was composed. After 250 responses were collected on a first-come, first-served basis, the survey was ended. The final analysis included 208 participants, excluding data from 42 who had insufficient responses.” (Page 2)
Point 2: In Measurements, Acceptance and Action Questionnaire-Stigma, the authors write that «The AAQ-S was translated into Korean using the translation-reverse translation process with the permission of the original developers». Here, and in the next questionnaires described, it would be useful to know if the authors performed transcultural adaptation, or if it was not required (and why, in this case).
Response 2: In response to the reviewer's comments, the following has been added to Procedure.
“The double translation method suggested by Waltz et al. [28] was used to translate the AAQ-S into Korean. One nursing professor and one nursing doctor, who were bilingual speakers of Korean and English, translated the AAQ-S into Korean, and revised and supplemented the translated version while comparing the results. Another doctor who spoke English as his mother tongue and was fluent in Korean translated the Korean version back into English. After the reverse translation, a nursing major fluent in English and Korean performed comparative analysis and verified whether there were any items with differences in meaning from the original tool. To check the content validity of the translated tool, two nursing professors and three diabetes experts verified the validity of the questionnaire. They also checked whether the contents of this questionnaire were applicable to Korean culture. The content validity index (CVI) evaluates the degree to which the tool is appropriate for the measurement of stigma acceptance behavior on a 4-point Likert scale ranging from 4=“strongly agree” to 1=“strongly disagree.” All items were confirmed to be valid with a CVI of 0.8 or higher. In addition, it was judged that the contents of the questionnaire can be applied to Korean culture. “ (Page 3)
Point 3: In the limitations section, perhaps the authors could make some kind of comment of the potential selection bias (if they consider that it could exist), and how it could affect their results and conclusions.
In think that if these aspects are better explained, the readers could better contextualize and better understand the results and the conclusions obtained. Nevertheless, I think that this is, overall, a very good research that can be useful for many researchers. Congratulations.
Response 3: We have added the following content to reflect the reviewers' comments.
“The significance of the AAQ-S-K is presented based on the validation results. First, this tool objectively measures the psychological flexibility of patients with diabetes by measuring their acceptance of stigmatizing thoughts. Second, it is possible to compare the AAQ-S-K with the AAQ-S translated into other languages. Third, this tool focuses on the emotional aspects of being diagnosed with diabetes, and can be applied in various studies in the future. Despite these advantages, there are some limitations to this study. First, the study sample was rather small. In addition, threr is a possibility that bias may have occurred because some participants were included. Therefore, care should be taken when generalizing the results of the study. Second, in the construct validity analysis, some of the fit indices did not meet the criteria. To address these limitations, further studies need to validate items using a larger study sample through random sampling. These studies will help to measure the acceptance of stigma thoughts in diabetes patients.” (Page 7)
